# Profile of Children with Undernutrition Admitted in Two Secondary-Level Hospitals in Maputo City, Mozambique

**DOI:** 10.3390/nu16071056

**Published:** 2024-04-04

**Authors:** Idalécia Cossa-Moiane, Clémentine Roucher, Maiza Campos-Ponce, Colleen Doak, Adilson Bauhofer, Assucênio Chissaque, António Prista, Nilsa de Deus, Katja Polman

**Affiliations:** 1Instituto Nacional de Saúde (INS), EN1, Marracuene 3943, Mozambique; adilson.bauhofer@ins.gov.mz (A.B.); assucenio.chissaque@ins.gov.mz (A.C.); nilsa.dedeus@ins.gov.mz (N.d.D.); 2Institute of Tropical Medicine, Nationalestraat 155, 2000 Antwerpen, Belgium; clementine.roucher@live.fr (C.R.); kpolman@itg.be (K.P.); 3Global Health Institute, Faculty of Medicine and Health Sciences, University of Antwerp, 2000 Antwerp, Belgium; 4Department of Health Sciences, VU University Amsterdam, Van der Boechorststraat 7, 1081 BT Amsterdam, The Netherlands; m.camposponce@vu.nl; 5Center for Health Sciences Education, College of Health Sciences, St. Ambrose University, 1320 W. Lombard Street, Davenport, IA 52803, USA; doakcolleenm@sau.edu; 6Instituto de Higiene e Medicina Tropical, Universidade Nova de Lisboa, 1349-008 Lisboa, Portugal; 7Faculty of Physical Education and Sports, Universidade Pedagógica, Maputo 1100, Mozambique; aprista1@gmail.com; 8Departamento de Ciências Biológicas, Universidade Eduardo Mondlane, Maputo 1100, Mozambique

**Keywords:** children, undernutrition, risk factors, anthropometry, intestinal parasites, double burden of malnutrition, Mozambique

## Abstract

Mozambique has one of the highest child undernutrition rates in Sub-Saharan Africa. The aim of this study was to characterize the profile of children from 1 to 14 years old hospitalized for undernutrition and to explore associated risk factors. Clinical, demographic, socioeconomic, and environmental data were collected. Anthropometric measurements and stool samples were collected from a child and their caretaker. The wealth index was determined using Principal Components Analysis. A total of 449 children and their caretakers were enrolled. The children had a median age of 1.0 year [IQR: 1.0–2.0], and 53.9% (242/449) were male. Most were admitted with severe undernutrition (35.7%, 159/449 kwashiorkor and 82.0%, 368/449 with −3SD Z-score indexes). The most common co-morbidities were HIV (30.0%, 120/400), diarrhea (20.0%; 80/400), and anemia (12.5%; 50/400). Among the caretakers, 9.5% (39/409) were underweight, 10.1% (40/397) were overweight, and 14.1% (56/397) were obese. Intestinal parasites were found in 24.8% (90/363) children and in 38.5% (77/200) caretakers. The majority of children (60.7%, 85/140) came from low- to middle-wealth households. Most were severely undernourished, suggesting that they seek medical care too late. The finding of overweight/obese caretakers in combination with undernourished children confirms that Mozambique is facing a double burden of malnutrition.

## 1. Introduction

Undernutrition is a significant public health concern with multiple negative consequences for individuals, communities, and society. Globally, in 2020, it was estimated that 149 million children under five years of age were stunted (too short for age) and 45 million wasted (too thin for height) [1]. In the same year, 21% of children under 5 years of age were wasted, and 27% were stunted only in Africa [2]. Mozambique has the fifth highest prevalence of children chronically undernourished in Africa, with 42% under five years of age affected [3]. According to the Demographic and Health Survey (DHS) 2022/23, the national prevalence of children under the age of five with stunting and severe stunting are 36.7% and 13.4%, respectively [4,5], suggesting a six-fold reduction in chronic undernutrition compared to the 2011 DHS [6]. Nevertheless, undernutrition remains a serious public health problem in the country.

In addition, Mozambique is a country severely affected by Human Immunodeficiency Virus (HIV). In 2015, a national survey reported that 2.0% of children aged 6 to 23 months tested positive with HIV [7]. A more recent national survey conducted between 2021 and 2022 showed 12.5% of adults (15 years of age or older) with HIV, although HIV data in children were not reported [8]. However, in selected groups, such as hospitalized children with diarrhea, HIV prevalence is even higher (at 7.0%). Furthermore, undernutrition is more likely in individuals living with HIV [8]. It is well known that HIV and undernutrition negatively reinforce each other; while HIV exacerbates undernutrition, undernutrition accelerates the progression of HIV [9,10]. According to the World Health Organization (WHO), an HIV-positive child needs to consume 10% more energy when asymptomatic and 50–100% more energy when symptomatic than healthy, uninfected children to maintain growth [10]. Being a tropical and low-income country, parasitic infection is endemic in Mozambique. A national epidemiological survey between 2005 and 2007 targeting schoolchildren (*n* = 83,331) [11] showed that 53.5% were infected with soil-transmitted helminths (STHs). Like HIV, intestinal parasite infections (IPIs) have a reciprocal interaction with undernutrition [12,13]. Hence, co-infections with HIV and/or IPIs can contribute significantly to the occurrence and severity of undernutrition in Mozambican children and vice versa, with severe consequences for health.

Since 2014/15, Mozambique has seen improvements in access to safe water, and according to the 2019/20 Household Budget Survey (HBS), the province and city of Maputo have the highest prevalence of households consuming water from safe sources [14]. However, the same survey also found that the percentage of households drinking safe water decreased from the wealthiest to the poorest [14]. This shows the inequality of access to water from safe sources, with the poorest households more likely to drink water from unsafe sources. In addition, low household income, poor personal and collective hygiene practices, and overcrowded living conditions have been shown to contribute to the development of undernutrition [6,15,16,17]. Finally, Mozambique is facing a nutrition transition in which both child stunting and maternal overweight/obesity are highly prevalent [6]. Overweight and obesity are increasingly common, with prevalence rates of more than 20.0% in women and/or caretakers [6,18,19]. According to the DHS of 2011, the nutritional status of the caretakers (mothers) is correlated with the nutritional status of their children [13]. However, evidence of the double burden of malnutrition is still scarce in Mozambique [20].

Most data related to undernutrition are focused on children under five years of age, and few explore the underlying determinants. It is important to better understand the epidemiology of pediatric cases of clinical undernutrition, including children over the age of five. Therefore, this study aims to characterize the profile of children from 1 to 14 years old hospitalized for undernutrition in Maputo City, Mozambique, and to identify demographic, clinical, socioeconomic, and environmental factors associated with their undernutrition, including the health status of their caretakers.

## 2. Materials and Methods

### 2.1. Study Design

A cross-sectional analysis was conducted in Maputo Province and City in southern Mozambique from January 2018 to March 2020. Participants were children between 1 and 14 years old residing in Maputo City or Maputo Province, admitted with undernutrition classified according to the national guidelines [21]. The children were admitted to the malnutrition unit of the pediatric ward at Hospital Geral José Macamo (HJM) and Hospital Geral de Mavalane (HGM). At admission, all caretakers provided informed consent for their own participation and that of their child. Additionally, informed assent was asked of children aged 12 to 14 years for inclusion in the study. Children were excluded if they i. lived outside Maputo City or Maputo Province, ii. had been admitted to the hospital with malnutrition during the past weeks, iii. refused to participate, iv. had serious clinical conditions (requiring immediate intensive or surgical care), v. caretakers were unwilling or unable to provide informed consent and/or there was no informed assent (children 12–14), and/or vi. the child was unable to comply with the study requirements. The informed consent of the caretaker included an agreement to collect and use their own data. A caretaker was defined as the legal guardian who takes care of the child and lives with the child (e.g., mother, aunt, grandmother, uncle, sister, stepmother). Clinical information was obtained for each child. Anthropometric measurements were taken, and stool samples were collected from each child as well as from their caretaker. No incentives were provided to study participants.

### 2.2. Hospital: Clinical Data

Clinical data were obtained from the clinical record and filled in by a medical doctor during the hospitalization of the child, namely socio-demographic data, body temperature, presence of other diseases (malaria, pneumonia, and others), HIV infection (and information on antiretroviral therapy), presence of anemia, vitamin A intake, and presence of edema in arms and/or legs. Clinical records were used to identify undernutrition in the form of kwashiorkor, marasmus, or marasmic-kwashiorkor as recorded by physicians at admission. Additional information such as history of vaccination, history of breastfeeding, and other co-morbidities was collected from the child’s vaccination card, from the clinical record, and/or asked from the caretaker.

### 2.3. Anthropometric Measurements

At inclusion in the study, anthropometric measurements were taken to assess the nutritional status of the children and their caretakers. Height, weight, mid-upper arm circumference (MUAC), waist circumference (WC), and four skinfolds (triceps, biceps, subscapular, and suprailiac) were measured by trained technicians according to standard techniques [22].

Height was measured using a portable infantometer (Seca 417) for toddlers/infants who were equal to or less than 2 years old and not able to stand. A portable stadiometer (Seca 213) was used for children who were older than 2 years and/or able to stand. Height was measured to the nearest 0.1 cm. Weight was measured using a portable digital balance (Seca 876) to the nearest 0.1 kg. Weight was taken with the caretaker for children who were not able to stand, using the tare function on the scale to subtract the caretaker’s weight. Height and weight measurements were taken in light clothing and without shoes. The Z-scores were determined for each participant using Anthro software version 3.2.2 (for children less than 5 years old) and AnthroPlus software version 1.0.4 (for children from 5 to 14 years old and caretakers < 19 years old) [22,23,24]. For children, Body Mass Index (BMI) for age (BAZ) was determined, and results were classified as thinness or severe thinness (Appendix A). Additionally, age- and sex-adjusted nutritional Z-scores for children were calculated. Height-for-age Z-scores (HAZs) were classified as stunted or severely stunted, and weight-for-height Z-scores (WHZs) as wasted or severely wasted (Appendix A) were determined according to the WHO growth standards [22,23,24]. Per WHO recommendations, children with edema were classified with severe acute malnutrition and considered less than −3 Standard Deviation (SD) for all Z-score weight-related indexes (BAZ and WHZ) [25]. Z-score values of HAZ < −6SD, WHZ < −5SD, and BAZ < −5SD were categorized as implausible values according to WHO guidelines and were not included in our analysis [23,24]. For adults (≥19 years old), BMI (weight (kg)/length or height (m^2^)) was calculated, and nutritional status was defined using BMI to categorize as underweight, normal, overweight, or obese, as described in Appendix A [23,24].

MUAC, WC, and four skinfolds (SKs) were assessed as a proxy of body composition [25]. MUAC and WC were measured using a tape (Seca 201) to the nearest 0.1 mm. Skinfolds were measured using a manual caliper (Harpender CE 0120) to the nearest 0.1 mm. Consecutive measurements were conducted as follows: (1) twice for height and weight for all participants, (2) twice for skinfolds for participants aged 1 to 12 years, and (3) three times for skinfolds for participants older than 12 years. Repetitive measurements were averaged.

### 2.4. Stool Sample Collection and Parasitological Examinations

Upon admission at the hospital, a single stool sample was collected from each child and caretaker and sent to the Laboratory of Parasitology of Instituto Nacional de Saúde (INS) for parasitological examination. The samples were analyzed by the Kato Katz (KK) method using a 41.7 mg template and by the formol-ether concentration (FEC) method as described elsewhere [26,27] for helminths and protozoans detection, respectively. Additionally, a thick smear with a modified Ziehl–Neelsen (mZN) stain was performed to detect intestinal opportunistic parasites such as *Cryptosporidium* sp., *Cyclospora cayetanensis,* and *Cystoisospora belli* [26,27]. The slides were examined microscopically for the presence of parasites.

An individual was considered positive if at least one parasite was observed by any of the diagnostic methods. Single infection was classified as the presence of only one parasite in the stool sample and mixed infection as the simultaneous presence of more than one parasite in the same stool sample [26,27]. All stool samples were collected, labeled, processed, and stored following laboratory standard procedures and Good Clinical and Laboratory Practice (GCLP). Each slide was read by two microscopists, and in case of discordant results, a third reading was performed. Results were reported based on an agreement between two of the three results. If the discrepancy persisted, the same sample was processed a second time to recheck the results, and if the discrepancies persisted, an interlaboratory comparison was performed. Results that were concordant between the laboratories were recorded.

### 2.5. Household Environment Data

During the household visits, socioeconomic data were collected by questionnaire. The questionnaire was written in Portuguese but read and explained in an understandable language for the participant by trained members of the study team. The questionnaire was built based on the Mozambican DHS indicators [6]. The wealth index of the households of undernourished children was determined using the questions and scoring from the DHS surveys [6,28,29]. The households were ranked by their score, and then the ranking was divided into five equal parts, from quintile one (lowest—poorest) to quintile five (highest—wealthiest) (Appendix A), based on the DHS comparative reports [28] for the categorization of assets. In addition, the number of household members, number of bedrooms in the house, type of toilet, source of water supply, and treatment (yes/no) of drinking water were recorded.

### 2.6. Data Analysis

Descriptive statistics were used to describe the socio-demographic, clinical, socioeconomic, and household environment data of undernourished children. Age stratification was defined to explore the anthropometric pattern by considering the number of intervals and aligned with other literature on this topic [30]. Cross-tabulation was used to analyze the relationship between two or more variables. Chi-square tests were used to assess the association between two categorical variables, or if requirements were not met, Fisher’s exact test was used. *t*-tests were used to compare mean values between groups.

For each household, a wealth index was calculated to determine their economic status. Households with missing values for any of the assets were removed from the calculation [29]. If the variable/asset was owned by more than 95% or by less than 5% of households, it was excluded from the analysis [29]. Principal component analysis (PCA) was then used to create the wealth index and household rankings [6,28,29]. The resulting wealth quintiles were classified as lowest, second, middle, fourth, and highest index, considering the lowest index as the least wealthy [6,28]. The wealth index was calculated in STATA (version 15.1).

We used R version 4.1.0 (Vienna, Austria) and STATA (version 15.1) to conduct the descriptive statistics as well as the inferential analysis; *p*-values lower than 5% were considered statistically significant.

### 2.7. Ethical Clearance

The protocol of the present study obtained ethical approval from the *Comité Nacional de Bioética em Saúde* (National Committee on Bioethics for Health, CNBS) in Mozambique (registration number IRB00002657, reference number 76/CNBS/2016), the Institutional Review Board (IRB) of the Institute of Tropical Medicine (ITM), Antwerp (reference number ITG 1180/17), and the University of Antwerp (UA), Belgium (registration number B300201733618, reference number 17/39/436). Individuals positive for IPIs were treated with appropriate drugs based on the national guidelines [31].

## 3. Results

### 3.1. Hospital: Clinical Data

During the study period, 688 children aged 1 year to 14 years were admitted with undernutrition, 51.0% (351/688) of which at Hospital Geral José Macamo (HJM) and 49.0% (337/688) at Hospital Geral de Mavalane (HGM). Figure 1 shows the flowchart of the recruitment process at both hospitals.

Among the enrolled children, 5.3% (24/449) were admitted more than once to the malnutrition unit of the hospital (Figure 1), and 3.8% (17/449) died during hospitalization.

The children had a median age of 1.0 years [IQR: 1.0–2.0], and 53.9% (242/449) were male. The children’s caretakers had a median age of 26 years [IQR: 22–33 years], and 96.4% were female. The majority of the caretakers were the child’s mother (92.0%; 413/449), and 80.6% were literate (358/444), meaning they had the ability to read and write and had been to school (Table 1).

The clinical records (Table 2) showed that 35.7% of the children had kwashiorkor (159/446), followed by marasmus (33.9%; 151/446) and a combination of both (22.9%; 102/446). The presence of edema in arms and/or legs was observed in 52.2% (226/433) of children. Of the 212 children for whom information was available, 30.7% (65/212) had received vitamin A.

Undernutrition associated infections/co-morbidities were observed in 89.5% (400/447) of the children, of which HIV (30.0%, 120/400), diarrhea (20.0%, 80/400), and anemia (12.5%, 50/400) were the most common.

### 3.2. Anthropometric Measurements of Children and Caretakers

The anthropometric characteristics of the children are presented in Table 3 and Table 4. There was no significant difference in height, weight, MUAC, waist circumference, and skinfolds (except for subscapular skinfolds in age group 1–4 years) between boys and girls from all age groups. Among caretakers, significant differences between males and females were observed in height and skinfolds (specifically triceps, biceps, subscapular, and suprailiac) (Table 3).

Most children in all age groups were admitted to the hospital with severe (−3SD) Z-score indexes (BAZ, WHZ, and HAZ). The majority of children were admitted with more than one condition related to undernutrition (93.7%, 369/394), meaning that they were admitted not only with one type of undernutrition according to the Z-score indexes (Table 4). Wasting (80.0%, 336/420) and thinness (51.3%, 106/207) were the most common nutritional conditions observed in our study population (Table 4). There was no significant difference in nutritional status between boys and girls, except for wasting (*p* < 0.05) in children aged 1 to 4 years. Regarding the nutritional status of the caretakers, 67.0% (274/409) had a normal BMI, and 33.0% (135/409) were malnourished (underweight, overweight, or obese) (Table 4). Underweight was more common in women (9.6%, 38/397) than in men (8.3%, 1/12). Additionally, only women were overweight (10.1%, 40/397) or obese (14.1%, 56/397).

### 3.3. Intestinal Parasite Infections of Children and Caretakers

Upon admission to the hospital, 80.8% (363/449) of the children and 44.5% of the caretakers (200/449) provided a stool sample. Intestinal parasite infections were detected in 24.8% (90/363) of the children’s stools and in 38.5% (77/200) of the caretakers’ stools.

Infection with *A. lumbricoides* (28.9%), with *Cryptosporidium* sp. (16.7%), and mixed infections with *A. lumbricoides* and *T. trichiura* (10.0%) were the most common in children. Among the caretakers, *Entamoeba coli* (16.9%) and *T. trichiura* (15.6%) infections and mixed infections with *A. lumbricoides* and *T. trichiura* (15.6%) (Figure 2) were the most common. Of the child/caretaker dyads with stool samples, 8.9% had IPIs in both the stools of the child and caretaker (17/191).

### 3.4. Household Environment Data

A follow-up visit was made to 142 households, and complete data on the household environment was collected from 98.6% of these (140/142) to create wealth index scores and quintiles. The size of these households varied from 2 to 19 members, with 6 as the median value. Moreover, 87.9% (123/140) of the households had one to three bedrooms in the house, and 70.4% (92/140) reported untreated drinking water, with 50.7% (71/140) having no flush toilet and 62.9% (88/140) sharing toilet facilities with neighbors.

The wealth indexes scoring allowed for a nearly even split into quintiles, with 20.7% (29) for the lowest/poorest, 18.6% (26) for the second, 21.4% (30) for the middle, 19.3% (27) for the fourth, and 20.0% (28) for the highest/wealthiest quintile. As expected, 60.7% of the children came from households with wealth indexes in the first to third quintile range. Figure 3 shows that, in all income groups, wasting was most prevalent, followed by stunting in all quintiles. Thinness, although less frequent, was more prevalent in quintiles 3–5 (5.7–6.4%) than in the first two quintiles (3.6% in both). There were no significant differences.

## 4. Discussion

The present study aimed to characterize the profile of undernourished children (from 1 to 14 years of age) admitted to the malnutrition unit of the pediatric ward in two urban hospitals in Maputo City and to explore factors associated with their undernutrition, including the health status of their caretakers. The results show high prevalences of both kwashiorkor and marasmus in children admitted to the malnutrition units in two Maputo City hospitals. Most notably, these rates of kwashiorkor and marasmus are even higher than those reported in children admitted with undernutrition nearly twenty years earlier. A retrospective study from 2001 of children admitted to the malnutrition unit of the national reference hospital, Hospital Central de Maputo (HCM) in Maputo City [32], reported 32.9% with kwashiorkor. In comparison, kwashiorkor was identified in 35.7% of these children admitted for undernutrition in 2018–2020. The prevalence of marasmus showed a similar pattern, reported as 28.4% in 2001 and 33.9% in these results. Given that the number of health facilities and health workers has increased in 20 years [31] and that overall care has improved [32], it is worrying that the percentage of children with severe undernutrition has not gone down but has even increased. Severe undernutrition can have major long-term consequences for children, such as cognitive deficits, poor school performance, poor motor development, reduced productivity, and high vulnerability to non-communicable diseases [33,34,35,36]. In addition, severe undernutrition has been reported to be responsible for a nine-fold increase in the risk of mortality [37,38] and is considered the most common condition related to the six causes of death in childhood mortality surveillance in sub-Saharan and Asian sites [39].

The high frequency of children with severe undernutrition, as reported in the clinical records, was confirmed by our anthropometric measurement data, highlighting the potential of anthropometry as a monitoring tool for undernourished children in a clinical setting. In short, we found 80.0% wasting and 51.3% thinness among hospitalized children with undernutrition, and more than a quarter of them were severely undernourished (<−3SD for all Z-score indexes). We also observed that a significant proportion of female caretakers were overweight (10.1%) or obese (14.1%), suggesting the presence of DBM within households. DBM is a topic that has not been extensively studied in Mozambique to date [4,6]. A comprehensive review of DBM and associated factors in low- and middle-income countries reported prevalences of 49.6% overweight and 17.9% obesity in Mozambique in 2011 [20]. Prevalence rates of DBM vary across the world [17,40,41]. Various determinants, including maternal age, mother’s height, education, occupation, dietary practices, breastfeeding, family income, family size, and urbanization type, have been reported to be linked to DBM [42]. As seen, several risk factors can be pointed out as associated with DBM occurrence worldwide (including Mozambique). Our results are consistent with other studies relating DBM to the nutritional transition [43], even in this population of children admitted to the hospital with clinical undernutrition. Further studies are needed to investigate the magnitude of the DBM in Mozambique and associated risk factors.

Recognizing the growing significance of DBM as a public health concern in LMICs, the World Health Organization (WHO) recommends five double-duty actions (DDAs): (i) promotion and protection of exclusive breastfeeding for the first six months and beyond; (ii) encouragement of appropriate early and complementary infant feeding; (iii) implementation of marketing regulations to mitigate misuse of breast-milk substitute; (iv) initiation of maternal nutrition and antenatal care programs; and (v) establishment of school feeding policies and programs [44]. These measures are being adopted globally, with Mozambique currently focusing on DDAs (i), (ii), and (iv) [45,46]. So far, these interventions have contributed to an increase in the cure rate of children with acute undernutrition in ambulatory care from 89.0% in 2020 to 91.0% in 2021 [47]. While all provinces of Mozambique have reached the established goal of 80% reduction in undernutrition, the city of Maputo has not [47], as reflected in our results.

Co-morbidities were common in our study population of undernourished children, with HIV (30.0%), diarrhea (20.0%), and anemia (12.5%) being the most frequent. This is in line with studies in hospitalized undernourished children in other African countries, such as South Africa and Tanzania [48,49]. These co-morbidities are well known to contribute to the development of undernutrition [48,49,50], not only separately but also together, as they are interrelated. For instance, HIV infection can lead to diarrhea, and diarrhea is associated with anemia [8,50,51,52,53,54]. While one-third of the children were HIV-positive, only 87.4% of them received antiretroviral therapy. Research is needed to better understand the barriers to receiving antiretroviral therapy and to understand caretakers’ awareness about the importance of testing and treatment to improve quality of life and to reduce undernutrition.

In our study, we also found 24.8% of the undernourished children to be co-infected with IPIs, the majority being STH, known risk factors for child undernutrition [15]. This finding is consistent with the literature showing undernutrition and IPIs to be closely linked, especially in children [1,25,54]. The 38.5% prevalence of IPIs observed in their caretakers suggests exposure to shared environmental risk factors, including inadequate hygiene, consumption of unsafe food and water, and potential exposure to substandard sanitation conditions [13,15]. Indeed, according to our data, the majority of households reported drinking untreated water (70.4%), not having a flush toilet (50.7%), and sharing a toilet with neighbors (62.9%).

Wealth inequality has been shown to play an important role in the development of undernutrition [15,54]. Indeed, we observed that undernourished children often came from poorer households. Interestingly, households in the middle-wealth quintiles in our study had the highest frequency of wasted children, while stunting was most frequent in the low-wealth quintiles (poor) households. Nevertheless, none of the differences by wealth index were statistically significant. However, the observed pattern differs from other studies, which have shown wasting and stunting to be more prevalent in the lowest wealth index households. In a study of Maputo street foods, highly processed foods were sold alongside natural foods with homemade dishes [55], indicating an early stage of the nutrition transition. Usually, in early nutrition transition contexts, there is a positive linear association between income and the consumption of highly processed snack foods [56,57]. In Maputo, it is unclear if household wealth influences the types of foods consumed. Further studies are needed to understand these patterns.

Our study had certain limitations, the first being the cross-sectional study design. The associations found cannot be used to establish causal relationships. Another limitation is the relatively small sample size of children with wealth index information. Despite these limitations, our data provide important novel insights into the characteristics of undernourished children who are hospitalized in Maputo City health facilities, not only under the age of 5 but up to the age of 14 (pediatric age for Mozambique). However, more studies are needed, for example, among non-hospitalized children or in rural areas, to see how far these characteristics also apply to other populations and settings in Mozambique. In addition, our study explored additional anthropometric data that are not routinely collected. The implementation of anthropometric proxies in a clinical setting can help physicians assess changes in nutritional status and body composition to better monitor hospitalized undernourished children during and after nutritional recovery [58,59,60]. To the best of our knowledge, this is the first study using skinfold data in undernourished children and their caretakers in Mozambique, thereby demonstrating the occurrence of DBM in households.

Malnutrition is a priority on the global agenda and is reflected in the Sustainable Development Goals (SDGs) in different ways [61]. Undernutrition itself directly impacts two SDGs and indirectly influences five others. Our study provides valuable data that can contribute to these SDGs and highlights the importance of taking a comprehensive approach to tackling undernutrition.

## 5. Conclusions

Our research describes severe undernutrition among hospitalized children aged 1 to 14 in Mozambique. A significant proportion of these children presented with co-morbidities and/or were infected with intestinal parasites.

The observed double burden of malnutrition in children and their caretakers underscores the critical need for nutritional interventions at the household level.

Further studies are needed i. to understand the increase in severe undernutrition despite improved healthcare, ii. to explore the application of anthropometric measurements in clinical settings for monitoring undernourished children before and after treatment and during nutritional recovery, and iii. to understand the magnitude, determinants, and implications of DBM.

## Figures and Tables

**Figure 1 nutrients-16-01056-f001:**
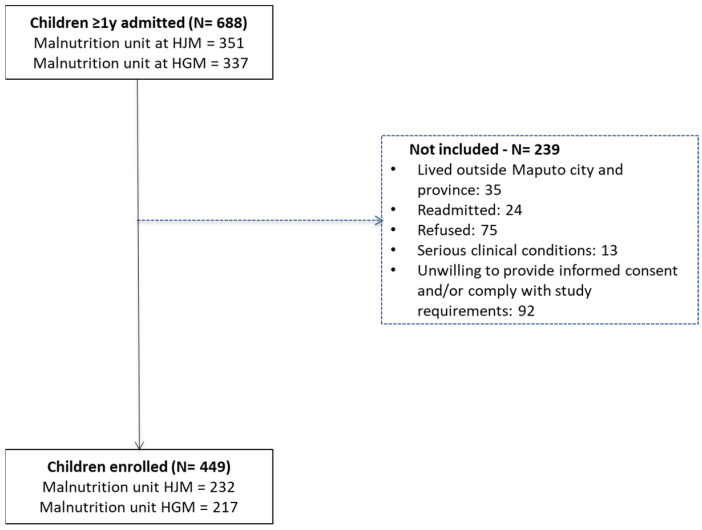
Flowchart of the recruitment process of undernourished children admitted at Hospital Geral José Macamo (HJM) and Hospital Geral de Mavalane (HGM) from January 2018 to March 2020.

**Figure 2 nutrients-16-01056-f002:**
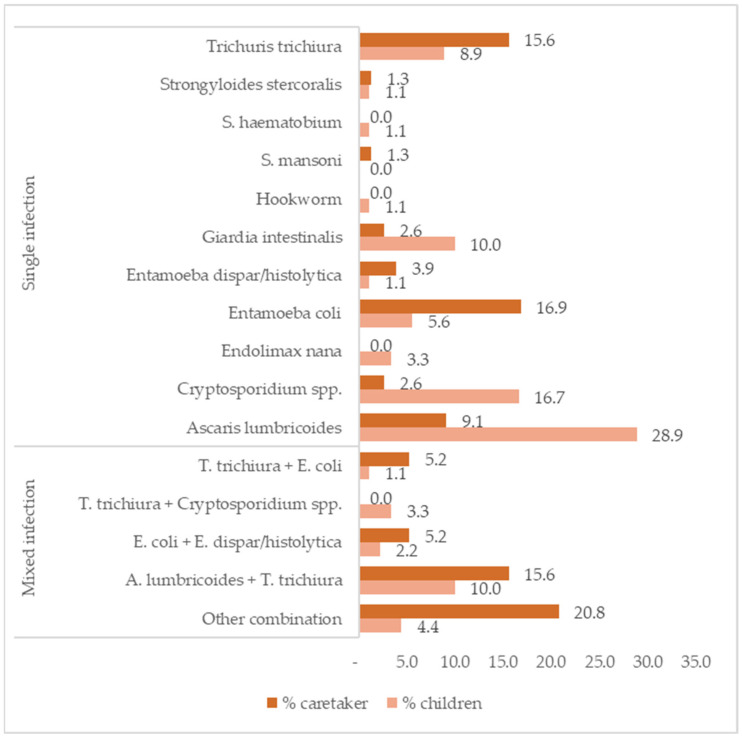
Frequencies of intestinal parasite infections by species and by single and mixed infection in children (N = 90) and caretakers (N = 77).

**Figure 3 nutrients-16-01056-f003:**
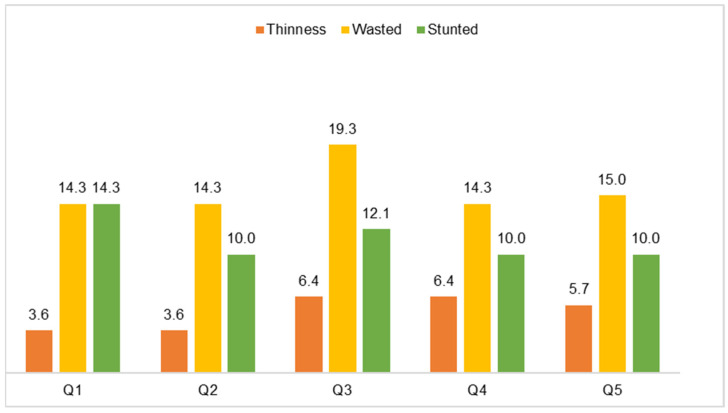
Proportion of children by nutritional status and wealth index quintile (N = 140).

**Table 1 nutrients-16-01056-t001:** Socio-demographic characteristics of the study population (N = 449).

Characteristics of Study Population	Children (N = 449)		Caretaker (N = 449)	
	*n*	%	*n*	%
Age in years, median [Q1–Q3]	1.0 [1.0–2.0]		26 (22–33)	
Sex				
Male	242	53.9	16	3.6
Female	207	46.1	433	96.4
Hospital				
HJM	232	51.7	NA	NA
HGM	217	48.3	NA	NA
Type of caretaker				
Mother	NA	NA	413	92.0
Father	NA	NA	15	3.3
Other	NA	NA	21	4.7
Literacy of caretaker (N = 444)				
Illiterate	NA	NA	86	19.4
Literate	NA	NA	358	80.6

*n*—numerator, NA—not applicable, %—percentage.

**Table 2 nutrients-16-01056-t002:** Clinical data of the undernourished children admitted at both HGM and HJM (N = 449).

Characteristics	*n*	%
Classification of undernutrition (N = 446)		
Kwashiorkor	159	35.7
Marasmic-kwashiorkor	102	22.9
Marasmus	151	33.9
Other	34	7.6
Edema (N = 433)		
No	207	47.8
Yes	226	52.2
Vaccination (N = 449)		
Not vaccinated	10	2.2
Vaccinated	439	97.8
Vitamin A intake (N = 212)		
No	147	69.3
Yes	65	30.7
Ever breastfed (N = 441)		
No	12	2.7
Yes	429	97.3
Presence of infections/co-morbidities (N = 447)		
No	47	10.5
Yes	400	89.5
Type of infections/co-morbidities (N = 400)		
Anemia	50	12.5
Bronchopneumonia	15	3.8
Diarrhea	80	20.0
HIV	120	30.0
Malaria	12	3.0
Other(s)	123	30.8
Antiretroviral therapy (N = 119)		
No	15	12.6
Yes	104	87.4

N—denominator; *n*—numerator

**Table 3 nutrients-16-01056-t003:** Anthropometric values among children and caretakers by sex and age group.

Measurement Mean (SD)	Children									Caretakers		
	1–4 Years (N = 422)			5–9 Years (N = 22)			10–14 Years (N = 5)			Male (*n* = 16)	Female (*n* = 433)	*p*-Value
	Boys (*n* = 227)	Girls (*n* = 195)	*p*-Value	Boys (*n* = 12)	Girls (*n* = 10)	*p*-Value	Boys (*n* = 3)	Girls (*n* = 2)	*p*-Value			
Height (N = 437 children; N = 410 caretaker)	74.5 ± 5.5	74.1 ± 8.2	0.65	104.7 ± 10.6	108.2 ± 15.3	0.58	111.5 ± 16.6	139 ± 0.0	NR	166.4 ± 6.1	156.9 ± 6.4	<0.01
Weight (N = 206 children; N = 411 caretaker)	7.6 ± 1.6	7.5 ± 2.3	0.18	14.0 ± 1.9	13.0 ± 3.0	0.40	20.4 ± 0.0	22.3 ± 0.0	0.99	59.6 ± 7.0	56.8 ± 11.5	0.40
Presence of edema * (52.2%, 226/433, children)	56.8 (125/220)	48.4 (90/186)	0.9 a	41.7 (5/12)	30.0 (3/10)	0.68 b	66.7 (2/3)	50.0 (1/2)	1.00b	NA	NA	NA
MUAC (N = 445 children; N = 408 caretaker)	11.9 ± 1.6	12.0 ± 1.5	0.71	12.5 ± 1.3	11.9 ± 1.0	0.24	12.8 ± 1.2	13.0 ± 2.3	0.94	28.4 ± 3.6	26.9 ± 3.9	0.21
WC (N = 443 children; N = 405 caretaker)	39.8 ± 4.2	39.3 ± 4.6	0.29	48.3 ± 5.5	44.4 ± 4.3	0.08	47.2 ± 10.6	45.1 ± 8.6	0.83	74.3 ± 7.3	75.8 ± 11.2	0.65
Skinfolds												
Triceps (N = 437 children; N = 407 caretaker)	6.1 ± 2.1	6.4 ± 2.2	0.17	4.2 ± 1.7	4.0 ± 1.5	0.80	5.8 ± 0.1	4.3 ± 3.5	0.46	7.5 ± 4.8	17.8 ± 7.2	<0.01
Biceps (N = 436 children; N = 406 caretaker)	5.6 ± 1.9	5.7 ± 2.0	0.33	3.5 ± 1.4	3.4 ± 0.9	0.75	4.3 ± 1.0	3.9 ± 3.2	0.81	6.1 ± 3.1	3.9 ± 6.1	0.04
Subscapular (N = 429 children; N = 404 caretaker)	4.1 ± 1.4	4.6 ± 2.0	0.00	3.2 ± 2.0	2.7 ± 0.7	0.52	3.0 ± 0.2	2.8 ± 0.3	0.41	7.6 ± 1.5	13.3 ± 7.3	0.01
Suprailiac (N = 412 children; N = 230 caretaker)	5.5 ± 2.5	5.9 ± 2.7	0.11	3.2 ± 1.0	3.2 ± 1.6	0.93	3.8 ± 0.4	3.4 ± 2.0	0.74	5.3 ± 2.5	7.6 ± 3.4	0.02

N—number of subjects, NA—not applicable, NR—not reported, SD—standard deviation. a—chi-square test, b—Fisher’s exact test, *—in percentage.

**Table 4 nutrients-16-01056-t004:** Nutritional status among children and caretakers by sex and age group.

Nutritional Status	Children									Caretakers		
	1–4 Years (N = 422)			5–9 Years (N = 22)			10–14 Years (N = 5)			Male (*n* = 16)	Female (*n* = 433)	*p*-Value
	Boys (*n* = 227)	Girls (*n* = 195)	*p*-Value	Boys (*n* = 12)	Girls (*n* = 10)	*p*-Value	Boys (*n* = 3)	Girls (*n* = 2)	*p*-Value			
BMI-for-age (BAZ) [Thinness—51.3% (106/207)]												
% (*n*) Normal (BAZ ≥ −2SD)	44.7 (42/94)	57.0 (57/100)	0.23	14.3 (1/7)	20.0 (1/5)	1.00	0.0 (0/1)	0.0 (0/0)	NR	NA	NA	NA
% (*n*) Thinness (−3SD ≤ BAZ < −2SD)	25.3 (24/94)	19.0 (19/100)		0.0 (0/7)	0.0 (0/5)		0.0 (0/1)	0.0 (0/0)		NA	NA	NA
% (*n*) Severe thinness (BAZ < −3SD)	29.8 (28/94)	24.0 (24/100)		85.7 (6/7)	80.0 (4/5)		100 (1/1)	0.0 (0/0)		NA	NA	NA
BMI [Malnutrition—33.0% (135/409)]												
% (*n*) Normal	NA	NA	NA	NA	NA	NA	NA	NA	NA	91.7 (11/12)	66.3 (263/397)	0.26
% (*n*) Underweight	NA	NA	NA	NA	NA	NA	NA	NA	NA	8.3 (1/12)	9.6 (38/397)	
% (*n*) Overweight	NA	NA	NA	NA	NA	NA	NA	NA	NA	0.0 (0/12)	10.1 (40/397)	
% (*n*) Obese	NA	NA	NA	NA	NA	NA	NA	NA	NA	0.0 (0/12)	14.1 (56/397)	
Wasting [Wasted—80.0% (336/420)]												
% (*n*) Normal (≥−2SD)	16.4 (36/219)	26.7 (50/187)	0.03	0.0 (0/7)	0.0 (0/5)	1.00	0.0 (0/2)	0.0 (0/2)	NR	NA	NA	NA
% (*n*) Wasted (−3DS ≤ WHZ < −2SD)	9.1 (20/219)	10.2 (19/187)		14.3 (1/7)	20.0 (1/5)		0.0 (0/2)	0.0 (0/2)		NA	NA	NA
% (*n*) Severe wasted (WHZ < −3SD)	74.3 (163/219)	63.1 (118/187)		85.7 (6/7)	80.0 (4/5)		100 (2/2)	100 (2/2)		NA	NA	NA
Stunting [Stunted—47.6% (195/410)]												
% (*n*) Normal (≥−2SD)	51.2 (111/217)	55.4 (102/184)	0.66	63.6 (7/11)	66.7 (4/6)	0.79	50.0 (1/2)	100 (1/1)	1.00	NA	NA	NA
% (*n*) Stunted (−3DS ≤ HAZ < −2SD)	19.8 (43/217)	19.0 (35/184)		18.2 (2/11)	0.0 (0/6)		50.0 (1/2)	0.0 (0/1)		NA	NA	NA
% (*n*) Severe stunted (HAZ < −3SD)	29.0 (63/217)	25.5 (47/184)		18.2 (2/11)	33.3 (2/6)		0.0 (0/2)	0.0 (0/1)		NA	NA	NA
% (*n*) More than one condition (BAZ, WHZ, HAZ) [93.7% (369/394)]	96.0 (192/200)	91.7 (154/168)	0.20	100 (11/11)	80.0 (8/10)	0.49	100 (3/3)	50.0 (1/2)	0.60	NA	NA	NA

NA—not applicable due to the age of the participants and the anthropometric tool used. NR—not reported.

## Data Availability

The data from this study are not publicly available due to restrictions present in the consent forms. The data presented in this study are available on request from the corresponding author.

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
