# Peer review of "Profile of Children with Undernutrition Admitted in Two Secondary-Level Hospitals in Maputo City, Mozambique"

_nutrients, 2024, doi:10.3390/nu16071056_

Round 1

Reviewer 1 Report

Comments and Suggestions for Authors

This is a very interesting study of 449 children involving undernutrition in Mozambique which examines various characteristics and risk factors.    This study is important. However, I have some concerns/ recommendations for this paper that could improve it further:

1. What is the research question?

2. There is no mention about the generalizability of the study.

3.  What is novel about this study?

4. Did participants receive incentives for enrolling in the study or no?

5. There are a lot of English language/editing errors e.g. line 29, 43, 54, 56, 93, 99, 101, 121, 227.

6. Positive HIV status was a significant finding, yet the discussion section did not contain any recommendations on how to address this problem.

7. The abstract mentions average age of 1 year but how does this reconcile with the children aged 12 to 14 years on line 99.  

Thank you for the opportunity to review this manuscript.

Comments on the Quality of English Language

There are a lot of English language/editing errors e.g. line 29, 43, 54, 56, 93, 99, 101, 121, 227.  E.g. line 25 should state "from children and their caretakers" instead of "from a child and their caretaker"

Line 29 should state ", which were the most common"

Line 42 should state "was estimated that 149 million..."

Line 43, remove "for"

Author Response

  1. What is the research question?

Response: Our study is a descriptive study, and our research question is “what is the profile of children (one to 14 years old) hospitalized for undernutrition in Mozambique, and what are the demographic, clinical, socioeconomic, and environmental factors associated with their undernutrition?” This is described in the Abstract (line 23 to 24) and in the last paragraph of the Introduction (line 94 to 98).

  1. There is no mention about the generalizability of the study.

Response: We are thankful for this comment, and we added the following sentence to the paragraph on limitations of the Discussion: “However, more studies are needed, for example among non-hospitalized children or in rural areas, to see in how far these characteristics also apply to other populations and settings in Mozambique.” (line 451 to 453).

  1. What is novel about this study?

Response: We believe that our study provides novel insights by (i) including children above five years old, (ii) including the child’s caretaker in the study which demonstrated the occurrence of DBM in households and (iii) incorporating skinfold measurements as an additional tool for monitoring body composition during recovery. These are now explicitly mentioned in the Discussion following the section on limitations. We also added that “To the best of our knowledge, this is the first study using skinfold data in undernourished children and their caretakers in Mozambique, thereby demonstrating the occurrence of DBM in households.” (line 464 to 466).

  1. Did participants receive incentives for enrolling in the study or no?

Response: We appreciate this question and would like to inform the reviewer that the participants did not receive any incentives for participation. We added a sentence on this in the Methods, at the end of section 2.1 Study design: “No incentives were provided to study participants.“ (line 125 to 126) 

  1. There are a lot of English language/editing errors e.g. line 29, 43, 54, 56, 93, 99, 101, 121, 227.

Response: We apologize for the English language/editing errors. We have now substantially revised the text with the help of a native speaker. See track changes throughout the manuscript. 

  1. Positive HIV status was a significant finding, yet the discussion section did not contain any recommendations on how to address this problem.

Response: We thank the reviewer for this comment and added the following sentence in the paragraph on co-morbidities in the Discussion: “While one third of the children were HIV positive, only 87.4% of them received antiretroviral therapy. Research is needed to better understand the barriers to receiving antiretroviral therapy and to understand caretakers’ awareness about the importance of testing and treatment to improve quality of life and to reduce the undernutrition.” (line 391 to 396)

  1. The abstract mentions average age of 1 year but how does this reconcile with the children aged 12 to 14 years on line 99.

Response: The study population consisted of children from one to 14 years old (which is the pediatric age range in Mozambique). To avoid misunderstanding on the age range, we adjusted the sentence with the children aged 12 to 14 years into: “At admission, all caretakers provided informed consent for their own participation and that of their child. Additionally, informed assent was asked of children aged 12 to 14 years for inclusion to the study.“ (line 111 to 114). 

Comments on the Quality of English Language

There are a lot of English language/editing errors e.g. line 29, 43, 54, 56, 93, 99, 101, 121, 227.  E.g. line 25 should state "from children and their caretakers" instead of "from a child and their caretaker"

Line 29 should state ", which were the most common"

Line 42 should state "was estimated that 149 million..."

Line 43, remove "for"

Response: We appreciate the recommendations on the quality of English language and substantially revised the text with the help of a native speaker. See track changes throughout the manuscript.  

Reviewer 2 Report

Comments and Suggestions for Authors

Thank you for the opportunity to review this heartbreaking but critically important paper describing indices of malnutrition, and associated risk factors, in 1 to 14 year old children hospitalized for malnutrition in Maputo City or Province, Mozambique. Your study design is outstanding, and the data collected thorough and clearly relevant to your study aims.

My questions concern 3 of your observations:

1. Since diarrhea is the second most common comorbidity behind HIV infection, and certainly contributes to malnutrition, I would suggest discussing the possible etiologies, of which there are clearly several (HIV?, intestinal parasites?, etc).

2. I might be most troubled by the fact that incidence of wasting is highest in the middle income (Q3) and highest income (Q5) quintiles. You mention this in paragraph 6 of your Discussion. I agree that further studies are required to address this observation, but can you speculate why this may be occurring?

3. This leads directly to my last question. The most common child/caregiver (mother) combination was a severely malnourished child with an overweight or obese mother. You discuss this in paragraph 3 of your Discussion, and refer the reader to other studies of DBM, but can you speculate on why this occurring? A similar study in India (Singh S, Shri N, Singh A. Inequalities in the prevalence of double burden of malnutrition among mother-child dyads in India. Sci Rep. 2023 Oct 7;13(1):16923. doi: 10.1038/s41598-023-43993-z. PMID: 37805548; PMCID: PMC10560231) reported that this was most common in wealthy households, with higher educated mothers, and attributed it in part to nutritional transition.

Author Response

  1. Since diarrhea is the second most common comorbidity behind HIV infection, and certainly contributes to malnutrition, I would suggest discussing the possible etiologies, of which there are clearly several (HIV?, intestinal parasites?, etc).

Response: We appreciate the reviewer’s comment and included the following sentences to the paragraph on co-morbidity in the Discussion: “These co-morbidities are well known to contribute to the development of undernutrition [1–3], not only separately but also together, as they are interrelated. For instance, HIV infection can lead to diarrhea and diarrhea is associated with anemia [3–6]” (line 387 to 391) 

  1. I might be most troubled by the fact that incidence of wasting is highest in the middle income (Q3) and highest income (Q5) quintiles. You mention this in paragraph 6 of your Discussion. I agree that further studies are required to address this observation, but can you speculate why this may be occurring?

Response: We were also quite puzzled by this finding. We speculate that in this urban setting, the higher frequency in the middle-income quintiles can be explained by the stage of the nutrition transition in Mozambique. In the early nutrition transition experience middle-income households have more access to snack foods because they can afford them while the lowest-income households cannot. Malnutrition from the consumption of nutrient poor, highly processed food (snacks) is consistent with the literature [7] and could explain our results in urban Maputo. We added the following sentences to paragraph 6 in the Discussion: “Interestingly, households in the middle-wealth quintiles in our study had the highest frequency of wasted children, while stunting was most frequent in the low-wealth quintiles (poor) households. Nevertheless, none of the differences by wealth index were statistically significant. However, the observed pattern differs from other studies which have shown wasting and stunting to be more prevalent in the lowest wealth index households. In a study of Maputo street foods, highly processed foods were sold alongside natural foods with homemade dishes [1], indicating an early stage of the nutrition transition. Usually, in early nutrition transition contexts, there is a positive linear association between income and the consumption of highly processed snack foods [2,3].” (line 419 to 431) 

  1. This leads directly to my last question. The most common child/caregiver (mother) combination was a severely malnourished child with an overweight or obese mother. You discuss this in paragraph 3 of your Discussion, and refer the reader to other studies of DBM, but can you speculate on why this occurring? A similar study in India (Singh S, Shri N, Singh A. Inequalities in the prevalence of double burden of malnutrition among mother-child dyads in India. Sci Rep. 2023 Oct 7;13(1):16923. doi: 10.1038/s41598-023-43993-z. PMID: 37805548; PMCID: PMC10560231) reported that this was most common in wealthy households, with higher educated mothers, and attributed it in part to nutritional transition.

Response: we thank the reviewer for this comment and for the reference provided. We added the following sentence to paragraph 3 of the Discussion and added the reference from Singh et al., 2023: “As seen, several risk factors can be pointed out as associated with DBM occurrence worldwide (including Mozambique). Our results are consistent with other studies relating DBM to the nutritional transition [11], even in this population of children admitted to the hospital with clinical undernutrition.”(line 365 to 370).

References 1:

  1. Gavhi, F.; Kuonza, L.; Musekiwa, A.; Motaze, N.V. Factors Associated with Mortality in Children under Five Years Old Hospitalized for Severe Acute Malnutrition in Limpopo Province, South Africa, 2014-2018: A Cross-Sectional Analytic Study. PLoS ONE 2020, 15, e0232838, doi:10.1371/journal.pone.0232838.
  2. Bashaka, P.J.; Sawe, H.R.; Mwafongo, V.; Mfinanga, J.A.; Runyon, M.S.; Murray, B.L. Undernourished Children Presenting to an Urban Emergency Department of a Tertiary Hospital in Tanzania: A Prospective Descriptive Study. BMC Pediatr. 2019, 19, 327, doi:10.1186/s12887-019-1706-1.
  3. Children and AIDS Available online: https://www.unicef.org/tanzania/what-wedo/hiv-aids (accessed on 1 February 2023).
  4. Sambo, J.; Cassocera, M.; Chissaque, A.; Bauhofer, A.F.L.; Roucher, C.; Chilaúle, J.; Cossa-Moiane, I.; Guimarães, E.L.; Manhique-Coutinho, L.; Anapakala, E.; et al. Characterizing Undernourished Children Under-Five Years Old with Diarrhoea in Mozambique: A Hospital Based Cross-Sectional Study, 2015-2019. Nutrients 2022, 14, 1164, doi:10.3390/nu14061164.
  5. Nhampossa, T.; Sigaúque, B.; Machevo, S.; Macete, E.; Alonso, P.; Bassat, Q.; Menéndez, C.; Fumadó, V. Severe Malnutrition among Children under the Age of 5 Years Admitted to a Rural District Hospital in Southern Mozambique. Public Health Nutr. 2013, 16, 1565–1574, doi:10.1017/S1368980013001080.
  6. Guerrant, R.; Walker, D.; Weller, P. Tropical Infectious Diseases: Principles, Pathogens and Practice; 3rd ed.; Saunders Elsevier: British, 2011; ISBN 978-1-43773777-6.
  7. Ali, M.S.; Kassahun, C.W.; Wubneh, C.A.; Mekonen, E.G.; Workneh, B.S. Determinants of Undernutrition among Private and Public Primary School Children: A Comparative Cross-Sectional Study toward Nutritional Transition in Northwest Ethiopia. Nutrition 2022, 96, 111575, doi:10.1016/j.nut.2021.111575.
  8. Sousa, S.; Gelormini, M.; Damasceno, A.; Lopes, S.A.; Maló, S.; Chongole, C.; Muholove, P.; Casal, S.; Pinho, O.; Moreira, P.; et al. Street Food in Maputo, Mozambique: Availability and Nutritional Value of Homemade Foods. Health 2019, 25, 37–46, doi:10.1177/0260106018816427.
  9. Hinnig, P. de F.; Monteiro, J.S.; de Assis, M.A.A.; Levy, R.B.; Peres, M.A.; Perazi, F.M.; Porporatti, A.L.; Canto, G.D.L. Dietary Patterns of Children and Adolescents from High, Medium and Low Human Development Countries and Associated Socioeconomic Factors: A Systematic Review. Nutrients 2018, 10, 436, doi:10.3390/nu10040436.
  10. Kanerva, N.; Wachira, L.J.; Uusi-Ranta, N.; Anono, E.L.; Walsh, H.M.; Erkkola, M.; Ochola, S.; Swindell, N.; Salmela, J.; Vepsäläinen, H.; et al. Wealth and Sedentary Time Are Associated With Dietary Patterns Among Preadolescents in Nairobi City, Kenya. Nutr. Educ. Behav. 2023, 55, 322–330, doi:10.1016/j.jneb.2023.02.001.
  11. Singh, S.; Shri, N.; Singh, A. Inequalities in the Prevalence of Double Burden of Malnutrition among Mother-Child Dyads in India. Sci. Rep. 2023, 13, 16923, doi:10.1038/s41598-023-43993-z. 

Reviewer 3 Report

Comments and Suggestions for Authors

This study describes the nutrition profile of children admitted to two hospitals in Maputo City, Mozambique, and their caregivers. The paper is well-written and clearly presents the study. I just have a few comments.

1)Table 2 uses the terms kwashiorkor, marasm (should be marasmus), kwashiorkor-marasmatic and other, yet does not define those. How was it determined which one of these groups described the child?

2)Tables 3 and 4 appear to be the same.

3)Page 9, line 2, was should be were. (“…scores indesex (BAZ,WHZ, HAZ). The majority of children was”)

4)Figure 3 presents quintiles based on wealth indices. I would disagree with the conclusion that thinness was observed most in the middle to rich wealth indexes households. There seems to be a similar percentage across all quintiles (14.3-15) except Q3 (19.3), which is higher.

5)The first paragraph of the discussion should be in the results section.

6)The first sentence of the conclusion uses the word alarming. I would suggest saying one out XX instead of the non-descript term alarming.

Comments on the Quality of English Language

no

Author Response

1)Table 2 uses the terms kwashiorkor, marasm (should be marasmus), kwashiorkor-marasmatic and other, yet does not define those. How was it determined which one of these groups described the child?

Response: We appreciate the reviewer’s comment and have corrected the word marasmus in the table 2 (page 8). We have added the following sentence to clarify that diagnoses of kwashiorkor, marasmus, and kwashiorkor-marasmatic were recorded directly from the clinical record, as determined by the attending physician: “Clinical records were used to identify undernutrition in the form of kwashiorkor, marasmus, kwashiorkor-marasmus, as recorded by physicians at admission.” Methods section (134 to 136).

2)Tables 3 and 4 appear to be the same.

Response: we thank the reviewer for this observation, in fact the table 4 was mistakenly added twice, the correct table 3 is now included (page 9).

3)Page 9, line 2, was should be were. (“…scores indesex (BAZ,WHZ, HAZ). The majority of children was”)

Response: We apologize for the English language/editing errors. We have now substantially revised the text with the help of a native speaker. See track changes throughout the manuscript. 

4)Figure 3 presents quintiles based on wealth indices. I would disagree with the conclusion that thinness was observed most in the middle to rich wealth indexes households. There seems to be a similar percentage across all quintiles (14.3-15) except Q3 (19.3), which is higher.

Response: we agree and for clarity we have added “There were no significant differences” (line 316 to 317). We also changed the wording to emphasize the point that it is the lack of a statistical difference by wealth index that is interesting. “However, the observed pattern differs from other studies which have shown wasting and stunting to be more prevalent in the lowest wealth index households. In a study of Maputo street foods, highly processed foods were sold alongside natural foods with homemade dishes [1], indicating an early stage of the nutrition transition. Usually, in early nutrition transition contexts, there is a positive linear association between income and the consumption of highly processed snack foods [2,3]. In Maputo, it is unclear if household wealth influences the types of foods consumed. Further studies are needed to understand these patterns.” (line 422 to 435)

5)The first paragraph of the discussion should be in the results section.

Response: Thank you for drawing this to our attention, much of the first paragraph was a repeat of content from the results. We have deleted this and started the discussion with a focus on the aim of the study and a focus on comparing these results to research from 2001. The comparison provides context for the discussion of secular trends as follows as follows: “The present study aimed to characterize the profile of undernourished children (from one to 14 years of age) admitted to the malnutrition unit of the pediatric ward in two urban hospitals in Maputo city and to explore factors associated with their undernutrition, including the health status of their caretakers. The results show high prevalences of both kwashiorkor and marasmus in children admitted to the malnutrition units in two Maputo city hospitals. Most notably, these rates of kwashiorkor and marasmus are even higher than those reported in children admitted with undernutrition nearly twenty years earlier. A retrospective study from 2001, in children admitted to the malnutrition unit of the national reference hospital, Hospital Central de Maputo (HCM) in Maputo city [4], reported 32.9% with kwashiorkor. In comparison, kwashiorkor was identified in 35.7% of these children, admitted for undernutrition in 2018-2020. The prevalence of marasmus showed a similar pattern, reported as 28.4% in the study from 2001 and 33.9% in the current study” (line 323 to 340)

6)The first sentence of the conclusion uses the word alarming. I would suggest saying one out XX instead of the non-descript term alarming.

Response: We agree with the reviewer. The word alarming no longer appears in the text.

References 2:

  1. Sousa, S.; Gelormini, M.; Damasceno, A.; Lopes, S.A.; Maló, S.; Chongole, C.; Muholove, P.; Casal, S.; Pinho, O.; Moreira, P.; et al. Street Food in Maputo, Mozambique: Availability and Nutritional Value of Homemade Foods. Nutr Health 2019, 25, 37–46, doi:10.1177/0260106018816427.
  2. Hinnig, P. de F.; Monteiro, J.S.; de Assis, M.A.A.; Levy, R.B.; Peres, M.A.; Perazi, F.M.; Porporatti, A.L.; Canto, G.D.L. Dietary Patterns of Children and Adolescents from High, Medium and Low Human Development Countries and Associated Socioeconomic Factors: A Systematic Review. Nutrients 2018, 10, 436, doi:10.3390/nu10040436.
  3. Kanerva, N.; Wachira, L.J.; Uusi-Ranta, N.; Anono, E.L.; Walsh, H.M.; Erkkola, M.; Ochola, S.; Swindell, N.; Salmela, J.; Vepsäläinen, H.; et al. Wealth and Sedentary Time Are Associated With Dietary Patterns Among Preadolescents in Nairobi City, Kenya. Journal of Nutrition Education and Behavior 2023, 55, 322–330, doi:10.1016/j.jneb.2023.02.001.
  4. Cartmell, E.; Natalal, H.; François, I.; Ferreira, M.H.; Grahnquist, L. Nutritional and Clinical Status of Children Admitted to the Malnutrition Ward, Maputo Central Hospital: A Comparison of Data from 2001 and 1983. J Trop Pediatr 2005, 51, 102–105, doi:10.1093/tropej/fmh088.

Round 2

Reviewer 1 Report

Comments and Suggestions for Authors

The authors addressed my concerns.